# Restoring small water bodies to improve lake and river water quality in China

Wangzheng Shen [1,2,3], Liang Zhang [1,3] ✉, Emily A. Ury [4,5], Sisi Li [1,3], Biqing Xia [1,3] & Nandita B. Basu [2,4,6] ✉

Climate change, population growth, and agricultural intensification are increasing nitrogen (N) inputs, while driving the loss of inland water bodies that filter excess N. However, the interplay between N inputs and water body dynamics, and its implications for water quality remain poorly understood. Analyzing data from 1995 to 2015 across China, here, we find a 71% reduction in the area of small ($<10^{4.5}$ m$^2$) water bodies (SWB), primarily in high-N-input agricultural regions. Preferential loss of SWBs, the most efficient nutrient filters, places 42% of China at high water quality risk due to increasing N inputs and declining SWB density. Currently, N removal by water bodies is 986 kilotonnes year$^{-1}$, but restoring 2.3 million hectares of SWB could increase removal by 21%, compared to just 5% for equivalent restoration of large water bodies. Targeted SWB restoration is crucial for improving water quality and mitigating N pollution in China.

Lentic systems, including lakes, reservoirs, wetlands, and ponds, are critical resources that play a major role in global hydrologic and biogeochemical cycles[1,2]. Water bodies are sensitive to land use change and anthropogenic activity and, as such, are important sentinels of global change[2,3]. The importance of protecting water resources is recognized worldwide and was recently affirmed by the United Nations as part of their Sustainable Development Goals[4]. However, climate change coupled with human-driven land use change imposes considerable uncertainty for the future of these resources. Previous studies have pointed out that the growth of agriculture in the 19th and 20th centuries led to widespread declines in wetland coverage worldwide, particularly affecting small wetlands that are susceptible to drainage and conversion into cropland[4–9]. Simultaneously, global changes have also contributed to the formation and expansion of lentic systems, such as the enlargement of lakes resulting from climate warming-induced glacial and snow-melt processes[10]. Gains and losses, particularly of small water bodies, have not been well documented, nor the consequences evaluated. Under the dual pressures of climate change and human activity, spatial and temporal changes in water bodies, including changes in size and distribution, are particularly important as water bodies are not only sentinels but also regulators of global change[2].

Lentic systems are also important sinks for aquatic contaminants, such as excess nutrients from farm fields[11–13]. The anoxic conditions found in these ecosystems promote denitrification and help to reduce nitrogen (N) contamination from excessive fertilizer use[14]. Given that China is the world's largest fertilizer-consuming country and has a high N surplus[15–18], the removal of N in urban and agricultural runoff by small wetlands is an ecosystem service of high value. Nitrogen pollution in China has contributed to widespread eutrophication and incidences of harmful algal blooms[18,19], threatening water security and causing significant economic losses[13,20,21]. Since joining the Ramsar Convention on Wetlands in 1992, China has advanced wetland conservation through efforts like the 2003 National Wetland Conservation Program, which aimed to protect 90% of natural wetlands and restore 1.4 million hectares[7]. At the 2022 Ramsar Conference (COP14), China proposed a resolution to protect and restore small wetlands, urging global action. However, the

[1]Key Laboratory for Environment and Disaster Monitoring and Evaluation of Hubei, Jianghan Plain-Honghu Lake Station for Wetland Ecosystem Research, Innovation Academy for Precision Measurement Science and Technology, Chinese Academy of Sciences, 430077 Wuhan, China. [2]Department of Civil and Environmental Engineering, University of Waterloo, Waterloo, ON N2L 3G1, Canada. [3]University of Chinese Academy of Sciences, 100049 Beijing, China. [4]Department of Earth and Environmental Sciences, University of Waterloo, Waterloo, ON N2L 3G1, Canada. [5]Environmental Defense Fund, New York, NY 10010, USA. [6]Water Institute, University of Waterloo, Waterloo, ON N2L 3G1, Canada. ✉e-mail: lzhang@apm.ac.cn; nandita.basu@uwaterloo.ca

environmental benefits of restoring small wetlands, compared to large ones, remain unclear.

While N pollution and inland water body dynamics have been studied extensively, there is a lack of landscape-scale analysis that links the trends in N pollution with the dynamics of inland water bodies[22,23]. Here, we use a high-resolution land use map to quantify changes in the dynamics of inland water bodies across China from 1995 to 2015. Temporal trends in water body dynamics are then combined with a gridded nitrogen input dataset to explore how changes in water body dynamics can alter landscape-scale N retention, and to develop restoration scenarios for water quality improvement.

## Results and discussion
### Dramatic decline in small water bodies across China
We found a 43% decrease in the number of water bodies across China between 1995 and 2015, while the total area covered by them increased by 3.8% over this timeframe (Fig. 1a). The sharp decline in the number of water bodies, despite an increase in area, is driven by the loss of smaller water bodies across the landscape (Fig. 1b). Specifically, we find that the loss rate of water bodies smaller than $10^{4.5}$ m$^2$ range between −44% and −19% (1995–2005) and −63% and −9% in the following decade. In contrast, there is no loss and even a slight gain in larger water bodies over these two decades (Fig. 1b). There has also been a decline in the total area covered by small water bodies (SWB), from 26% between 1995 and 2005 to 23% in the following decade (Fig. 1c). While the area covered by small water bodies have decreased, the area covered by larger water bodies have increased slightly (Fig. 1c), contributing to an overall slight increase in the area of all water bodies. This breakpoint in the relationship at $10^{4.5}$ m$^2$ is interesting and conforms with the definition of small water bodies by Chinese national

standard (GB/T 42481-2023)[24] (hereafter referred to as small water bodies).

The preferential loss of small water bodies that is offset by the gain in area of larger water bodies has changed the size-frequency distribution of water bodies[25], from the more fractal patterns indicative of natural landscapes to more anthropogenic landscapes characterized by a greater proportion of larger water bodies[5,26,27]. We used the breakpoint of $10^{4.5}$ m$^2$ (Fig. 1b) to develop a power function relationship between water body size and frequency for water bodies larger than $10^{4.5}$ m$^2$ at the national scale (Fig. 1d). The forms of the power function are slightly different across the country (Supplementary Fig. 1); however, the ubiquity of the relationships demonstrates a pervasive pattern of loss of small water bodies (Supplementary Fig. 1). Our findings of the pervasive loss of smaller water bodies are parallel to other studies in North America[5,8,26,28,29], although the rate of loss in the more recent times is considerably higher in China. Of course, while focusing on the recent decades highlights the loss of small water bodies, analysis of wetland loss since the 1700 s notes that we have lost both small and large wetlands[4].

### Spatial pattern of small water body loss
The decrease in the number (Fig. 2a, b) and total area (Fig. 2c, d) of small water bodies is widespread across the country, with greater decreases in the last decade. Local gains in small water bodies are apparent in the coastal areas, and this can be attributed to the construction of stormwater ponds in these coastal cities to protect against flooding[30–32]. Land cover data over this timeframe highlight a dominance of SWB loss in agricultural areas, with 49% of the water bodies converted to cropland, 19% to forest, 18% to open grassland, and 7% to developed land, 4% to saline-alkali land and 3% to other land use

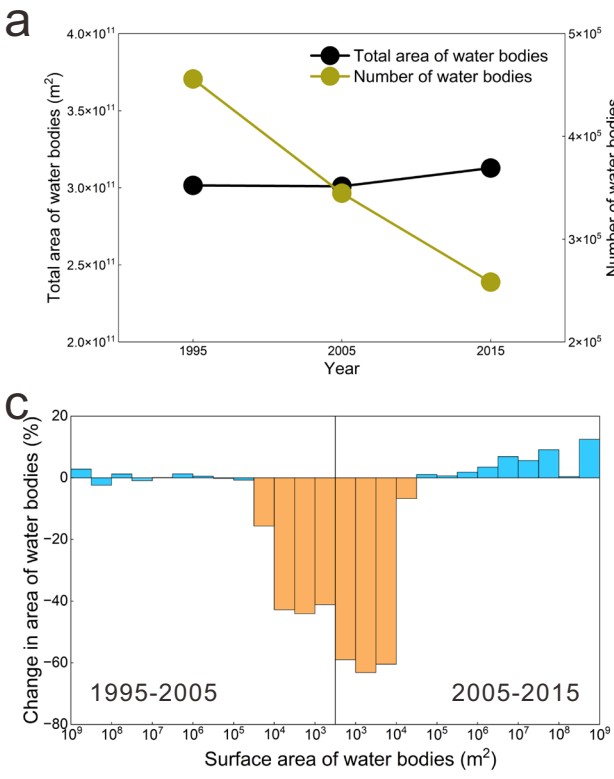

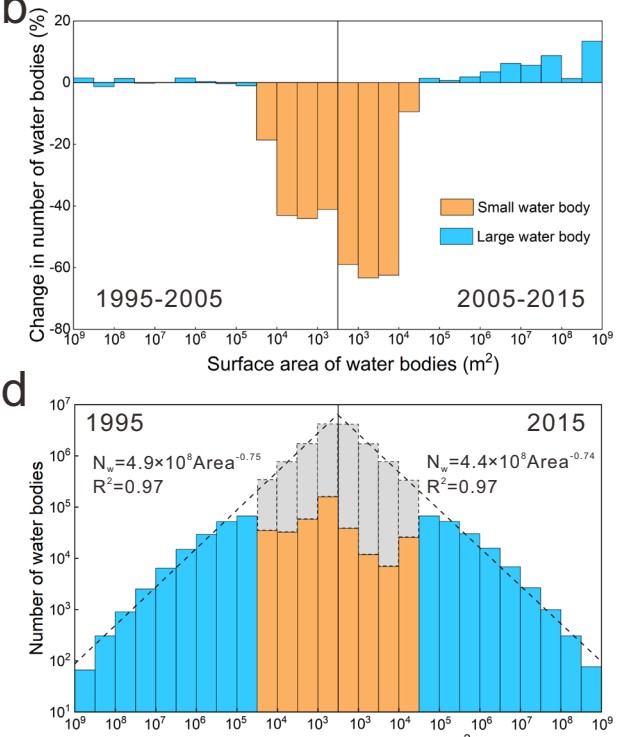

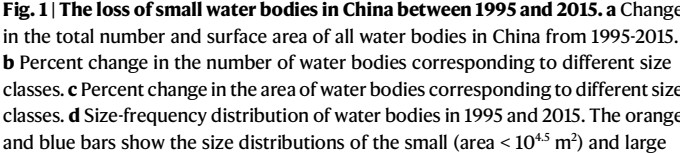

**Fig. 1 | The loss of small water bodies in China between 1995 and 2015. a** Change in the total number and surface area of all water bodies in China from 1995-2015. **b** Percent change in the number of water bodies corresponding to different size classes. **c** Percent change in the area of water bodies corresponding to different size classes. **d** Size-frequency distribution of water bodies in 1995 and 2015. The orange and blue bars show the size distributions of the small (area < $10^{4.5}$ m$^2$) and large water bodies. The dashed lines represent the results of fitting a power function using only large water bodies. The gray bars represent the expected distribution of small water bodies and their potential loss prior to 1995 or 2015. The $N_w$ in the equation represents the number of water bodies, Area in the equation represents the surface area of water bodies. Source data are provided as a Source Data file.

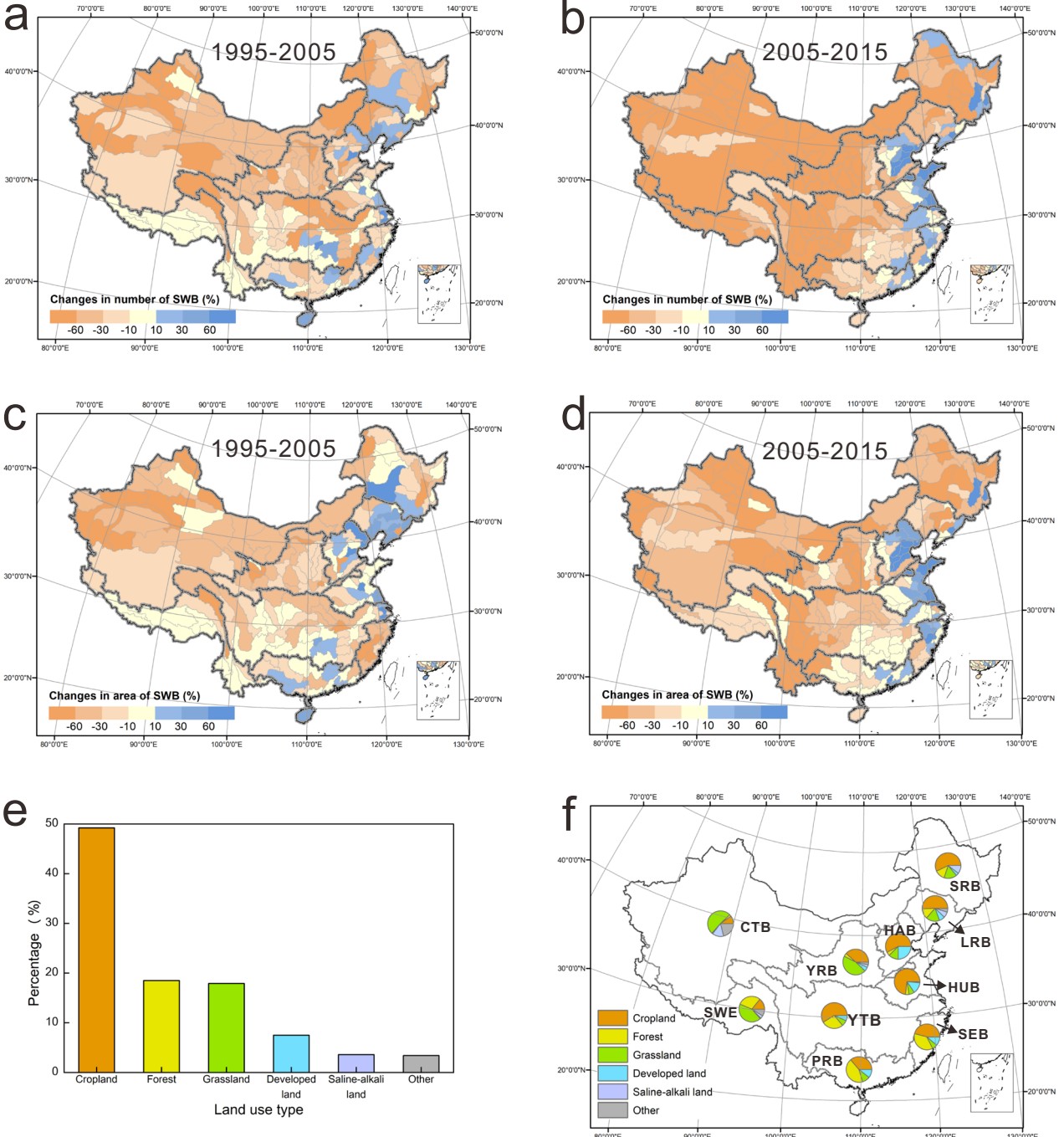

**Fig. 2 | Spatial patterns of small water body change across China.** Percent change in the number of small water bodies between (**a**) 1995 to 2005, and (**b**) 2005 to 2015. Percent change in area of small water bodies between (**c**) 1995 to 2005, and (**d**) 2005 to 2015. **e** Conversion pathways for small water bodies across China over the past two decades. **f** Small water bodies are converted primarily to croplands (orange in pie graphs) across eastern and central China and to grasslands (yellow in pie graphs) in the West. The thick gray lines demarcate the boundaries for the first-order watersheds: Songhua River Basin (SRB), Liaohe River Basin (LRB), Haihe River Basin (HAB), Yellow River Basin (YRB), Huaihe River Basin (HUB), Yangtze River Basin (YTB), Southeast Basin (SEB), Pearl River Basin (PRB), Southwest Basin (SWE), Continental Basin (CTB). Source data are provided as a Source Data file.

(Fig. 2e). The conversion to grassland is more common in western and central China, in the Continental, Southwest and the Yellow River Basins, while the conversion to cropland is more common in the agriculture dominated eastern and central Songhua, Liaohe, Haihe, Huaihe and Yangtze River Basins (Fig. 2f). Conversion to developed land occurs in the coastal Pearl, Southeast, Haihe, and the Huaihe River Basins (Fig. 2f). The conversion can be attributed to a combination of human activities like agricultural expansion or urbanization[4,7,33] and climatic factors, such as increasing frequency of droughts[34,35].

## Small water bodies are lost where they are needed the most

The preferential conversion of small water bodies to cropland has consequences for water quality, given croplands have higher nutrient inputs compared to other land use types[17,36], and small water bodies have higher rates of nutrient removal relative to their size[1,11,13,37]. Thus, we are losing small water bodies in regions where they are needed the most for protecting downstream waters.

To identify high-risk areas for water quality degradation, we used a recently developed map of grid-scale, landscape nitrogen surplus

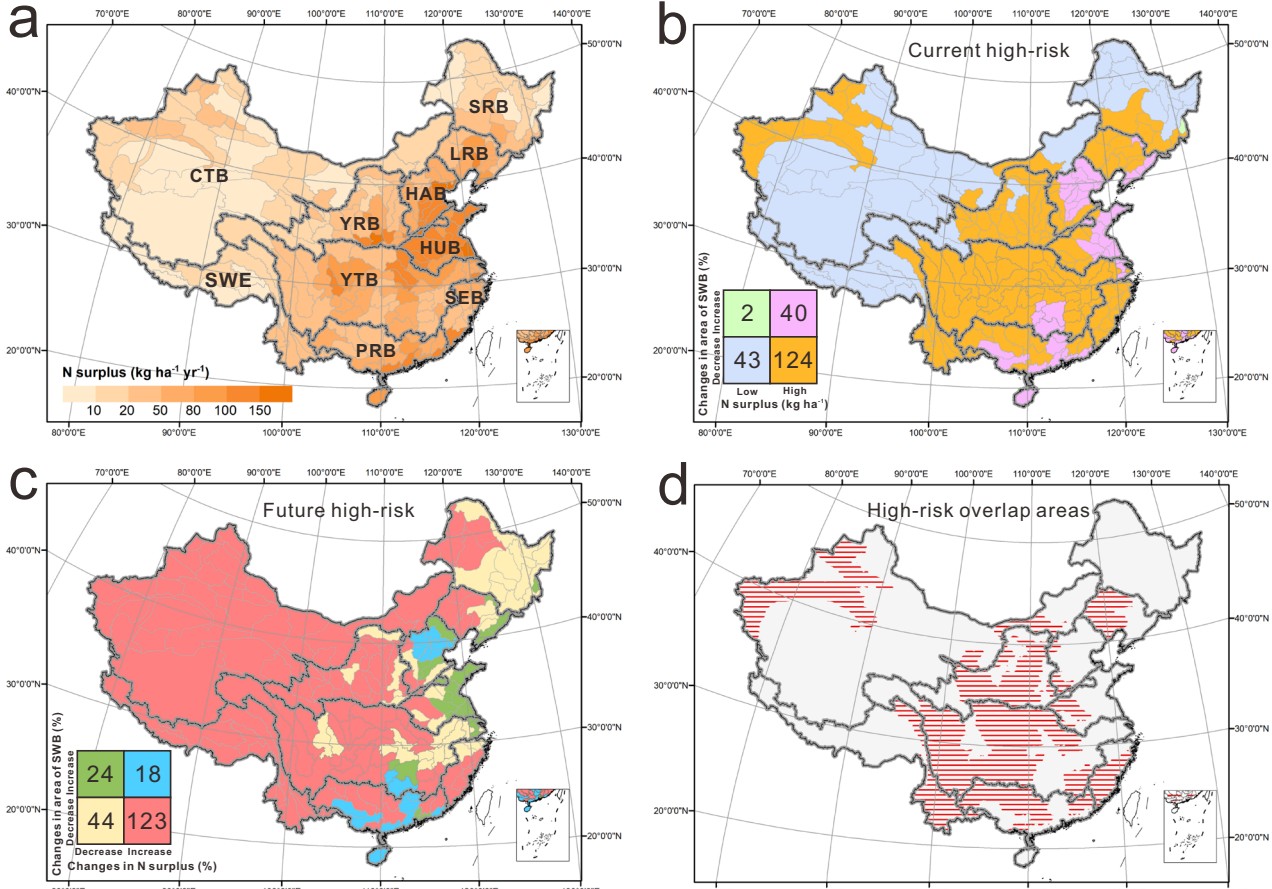

**Fig. 3 | Spatial co-occurrence of water body loss and N surplus across China.**
**a** Magnitudes of nitrogen surplus (kg ha⁻¹ yr⁻¹) across the third-order river basins in China (**b**) Relationship between 2015 N surplus and percent change in the area of small water bodies (SWB) between 1995 and 2015, highlighting current high-risk regions (orange region) with high N surplus (>17 kg ha⁻¹ yr⁻¹) and loss of SWB (**c**) Relationship between the percent change in N surplus and the percent change in SWB area, highlighting future high-risk regions where N surplus is increasing while water bodies are being lost (red region). **d** Regions that are both currently at high-

risk and under risk in the future (red regions), are identified as the intersection of current and future high-risk regions. The numbers in the squares correspond to the number of third-order river basins matching the respective colors. The thick gray lines demarcate the boundaries for the first-order watersheds: Songhua River Basin (SRB), Liaohe River Basin (LRB), Haihe River Basin (HAB), Yellow River Basin (YRB), Huaihe River Basin (HUB), Yangtze River Basin (YTB), Southeast Basin (SEB), Pearl River Basin (PRB), Southwest Basin (SWE), Continental Basin (CTB). Source data are provided as a Source Data file.

across China (Fig. 3b). Nitrogen (N) surplus magnitudes, estimated as the difference between anthropogenic N inputs (including fertilizer, manure, atmospheric deposition, biological N fixation, and straw return to the field) and crop N production, is an indicator of the excess N that contributes to water pollution[18,38,39]. High N surplus magnitudes are apparent in eastern and central China (Fig. 3a). This is also a region where a large number of small water bodies are being lost in the last two decades (Fig. 2).

We use the temporal trends in our N surplus and small water body datasets to identify areas that might be at greater risk to nitrogen pollution. We defined current high-risk regions as those with a high N surplus and where small bodies are being drained in the last two decades – these are intensive agricultural regions where wetlands are needed to tackle the high N surplus. We further define future high-risk areas as those where the N surplus is increasing while SWB are decreasing from 1995 to 2015. These are potentially less agriculturally intensive regions, where agricultural activities and thus N surplus is increasing, while small water bodies are being lost. Their N surplus magnitudes might still be low, but the increasing trend of N sources, with concomitant decrease in small water bodies create a high risk of future water quality deterioration. We identified 60% of the third-order river basins as current high-risk regions (orange region in Fig. 3b) with high N surplus (>17 kg ha⁻¹ yr⁻¹; global average N surplus[40]) and SWB

loss. The majority of regions in the Yangtze and Yellow River Basins are designated as current high-risk areas according to this metric. Nitrogen discharge threshold for surface water quality was exceeded in these regions as early as the 1980s (refs. 41,42). Despite considerable investment in addressing water pollution, nitrogen concentrations in surface water remain high[21]. We also identified future high-risk regions (red region in Fig. 3c) as areas where N surplus is increasing accompanied by a loss of small water bodies. This category encompasses approximately 78% of the third-order river basins, including regions in western China experiencing cropland expansion (Supplementary Fig. 2). Although water quality issues in these regions may not be currently as acute, they are at heightened risk in the future, with some early signs of increasing pollution in previously pristine lakes[43].

Perhaps most interesting is the intersection between the current and future high-risk areas (hatched regions in Fig. 3d) that encompass 42% of the third-order river basins. These areas in the Yangtze and the Yellow River Basin are of particular concern since they are already polluted, and pollutant sources (N surplus) are increasing while the small water bodies that filter the pollutants are decreasing. Despite significant investments made by the Chinese government in curbing water pollution[21,44], reversing the trend of water quality degradation is challenging, and requires more sustained efforts.

## Nitrogen removal by current water bodies

To further quantify the role of the inland water bodies in N removal, we estimated the mass of N removed as the product of the N removal potential, and the nitrogen inputs estimated from the N surplus dataset[45]. Nitrogen removal potential for each water body was estimated as a function of their size and geographic location, using an empirical model developed by ref. 46, based on measured data from 417 water bodies across China (see Methods). The actual nitrogen removed by each of the 258,237 water bodies was then estimated as a function of the water-body specific N-removal potential and the corresponding N surplus of the watershed where the water body is located. The water body specific estimates were then used in conjunction with the distributions of water bodies in each of the third-order river basins to estimate watershed-scale N removal rates (Fig. 4a).

We found water bodies across China to remove $986 \pm 142$ kilotonnes of N yr$^{-1}$, which represent 3% of the current landscape N surplus. Our numbers are consistent with an independent estimate of denitrification magnitudes (1100 kilotonnes yr$^{-1}$) in aquatic ecosystems across China[47], as well as a country-scale estimate of N removal across US ($860 \pm 160$ kilotonnes yr$^{-1}$)[36]. Watershed scale N removal rates varied across the basin (median: 0.5 kg ha$^{-1}$ yr$^{-1}$), and were similar to a US scale study on N removal in wetlands[36]. The highest removal rates were apparent in the agriculture-dominated Huaihe and Haihe river

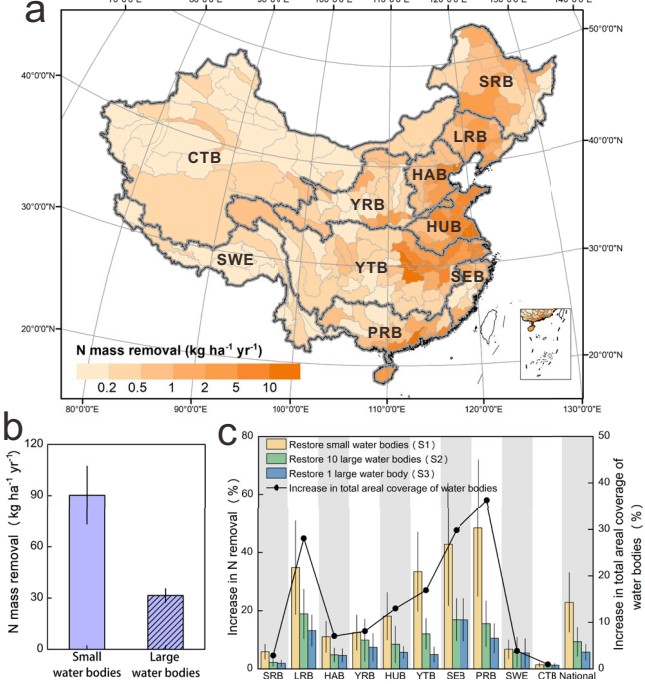

**Fig. 4 | Nitrogen removal in current water bodies and restoration scenarios.**
**a** Estimated nitrogen mass removal (kg ha$^{-1}$ yr$^{-1}$) by all water bodies aggregated to the third-order river basin scale and normalized by the corresponding basin area.
**b** Nitrogen mass removal by all small and large water bodies (kg ha$^{-1}$ yr$^{-1}$) across China, normalized by the corresponding water body area. The bars represent the median values and the whiskers the interquartile range (IQR). **c** Percent increase in N removal under different restoration scenarios, and the corresponding percent increase in area of water bodies which is the same across all three scenarios. Results highlight the increased N removal when restoring small water bodies with same areal coverage as larger water bodies. The error bars represent the IQR. The thick gray lines demarcate the boundaries for the first-order watersheds: Songhua River Basin (SRB), Liaohe River Basin (LRB), Haihe River Basin (HAB), Yellow River Basin (YRB), Huaihe River Basin (HUB), Yangtze River Basin (YTB), Southeast Basin (SEB), Pearl River Basin (PRB), Southwest Basin (SWE), Continental Basin (CTB). Source data are provided as a Source Data file.

basins which have the highest N inputs, as well as greater densities of small water bodies (Fig. 4a and Supplementary Fig. 3).

## Restoration of small water bodies to address nitrogen pollution

The persistent water pollution challenges in China has led to increased interest in restoring small water bodies that retain nutrients and mitigate downstream N loads[37,48]. However, it is not often clear which water bodies to restore and where in the landscape. Our national scale model highlighted that N removed by water bodies were strongly driven by size, such that areal N removal rate of small water bodies (90 kg ha$^{-1}$ yr$^{-1}$) is three times the removal rate of water bodies larger than $10^{4.5}$ m$^2$ (32 kg ha$^{-1}$ yr$^{-1}$) (Fig. 4b). The higher areal N removal rates of smaller water bodies occur due to their greater reactive surface area compared to volume that provides greater opportunities for biogeochemical reactions[36]. Thus, restoration scenarios that focus on restoring small water bodies have a strong potential to increase water quality benefits.

To test this hypothesis, we simulated three restoration scenarios using our national scale model (Fig. 4c). Under the first scenario (restoration of small water bodies, S1), small water bodies are restored to achieve a size distribution that mimics the original fractal distribution on the landscape. The results were compared to scenarios where the same area is restored, but through the restoration of 10 larger water bodies (S2) and one very large water body (S3) (see methods).

When analyzed at the country-scale, we found that restoration of SWB to mimic the original fractal landscape would lead to a 7% increase in the overall area of water bodies (2.3 million hectares), which closely aligns with the wetland restoration goal of the National Wetland Conservation Program for 2030 (1.4 million hectares)[49]. Such restoration could contribute to a $21 \pm 10\%$ increase in N removal ($211 \pm 94$ kilotonnes yr$^{-1}$) across the country, while restoring the same area as a single large water body would contribute to only a $5 \pm 3\%$ increase. The 7% increase in wetland area for our three scenarios is of course a function of the threshold of $10^{4.5}$ m$^2$ selected for the study. Traditional wetland conservation often prioritizes large wetlands, with the 82 Ramsar-listed wetlands in China averaging 100,000 hectares. However, our simulation results underscore the vital role of small wetlands in pollution control. This finding strongly supports China's National Wetland Conservation Program, the fulfillment of responsibilities by the Ramsar Convention on Wetlands' Contracting Parties worldwide, and the pursuit of Sustainable Development Goals through effective wetland restoration. While a different threshold value would lead to a different value for the area restored, our main finding that restoring smaller water bodies would have much greater water quality benefits would still be valid.

When analyzed at the river basin-scale, we found the percent increase in area required to mimic a fractal landscape is variable across the ten first-order river basins, ranging from a 1–3% increase in the northern Continental and Songhua River Basins to a 36% increase in the southern Pearl River Basin. The differences can be attributed to the differences in the initial size-frequency distributions for each of the ten basins (Supplementary Fig. 1). However, for all river basins restoration of small water bodies (Scenario 1) contributed to much greater N removal than restoration of larger water bodies. The differences are greatest in the highly polluted Yangtze River Basin, where restoring small water bodies can contribute to a $33 \pm 14\%$ increase in N retention, while restoring one large water body of the same area would contribute only a $5 \pm 3\%$ increase. These results suggest that when designing water body restoration strategies, the restoration of small water bodies should be given a higher priority.

To demonstrate the potential economic value of small water body restoration, we situate our findings in the context of conventional wastewater treatment costs[50,51]. To achieve N removal of a magnitude comparable to that provided freely as ecosystem services by our existing water bodies, it would cost US$8 $\pm$ 1 billion in 2015. As a point

of reference, it's worth noting that in 2022 China has allocated a budget of US$3.9 billion for water pollution prevention and control[50]. Reduction in N export due to the restoration of small water bodies could alleviate stress on existing wastewater treatment infrastructure. In fact, it would require 800 new wastewater treatment plants (WWTPs) to achieve the same N load reductions as is achievable in S1 where SWB are restored to mimic the fractal landscape[50]. The cost associated with these small water body losses is therefore US$2 ± 1 billion in China. The real value of water bodies comes from their ability to effectively reduce N at low influent levels, given WWTPs typically receive N at much higher concentrations and treating low-concentration wastewater is much more costly[50]. Thus, the true economic loss associated with the disappearance of small water bodies is much greater and not feasible to replace with existing technology.

Our research underscores the importance of protecting and restoring small water bodies as vital components of the landscape, not only for their role in global hydrologic and biogeochemical cycles but also for their significant contribution to water quality improvement. With water pollution being a pressing issue in China, the restoration of small water bodies offers a cost-effective and environmentally sustainable solution, alleviating stress on existing wastewater treatment infrastructure. It is imperative that efforts are made to prioritize the preservation and restoration of these small water bodies to safeguard the nation's water resources and ensure a sustainable and healthy environment for future generations.

## Methods

In the following, we provide details on our datasets, analytical methods and modeling approach for estimating N removal by water bodies, scenario analysis of wetlands restoration, and associated economic estimates.

### Developing size-dependent trends in water body dynamics across China

We used the Chinese Multi-Period Land Use and Land Cover Remote Sensing Monitoring Data Set (CNLUCC, 30-m resolution) to identify water bodies in 1995, 2005, and 2015 across the ten major first-order river basins in mainland China (Supplementary Table 1, Supplementary Table 2). The CNLUCC dataset, generated using remote sensing imagery (Landsat TM/ETM or Landsat 8), is provided by the Data Center for Resources and Environmental Sciences, Chinese Academy of Sciences[52]. The CNLUCC data has been extensively used in previous studies (such as refs. 53,54), and its accuracy has been confirmed through multi-year field surveys[55]. Although higher resolution land use can be identified with the development of remote sensing satellite technology in recent years[56], CNLUCC data is the most comprehensive land use data we can obtain considering the time span (two decades) and spatial scale (national-scale) of interest in this study. The annual precipitation for these years closely aligns with the long-term average precipitation (Supplementary Table Fig. 4). The variation in precipitation across these three years is less than 3%, ensuring data comparability.

Water bodies identified in this paper include lakes, reservoirs, ponds, and wetlands, but exclude rivers and paddy fields. Water body loss analysis was done at the country-scale (Fig. 1), as well as across 10 first-order and 207 third-order river basins (Fig. 2). Water bodies were grouped within different size bins ($10^{2.5}–10^3$ m$^2$, $10^3–10^{3.5}$ m$^2$, etc.), and the percent loss in water bodies between 1995–2005 and 2005–2015 was estimated within these bins, by comparing the distributions of water bodies between the years 1995, 2005, and 2015. We further defined water bodies smaller than $10^{4.5}$ m$^2$ as small water bodies (SWB), which conforms with the Chinese national standard "Specification for conservation and management of small wetlands" (GB/T 42481-2023)[24]. We then fitted a power function relationship to water bodies larger than this threshold, following the assumption that preferential loss of small water bodies creates deviations from the theoretical

power function relationships between the size of the water bodies and their abundance across the landscape[5] ($N_w = cA^d$; $A$ in m$^2$ is the surface area of the water body, $N_w$ is the number of water bodies corresponding to the size, and $c$ and $d$ are empirical constants). The power-law relationship is considered to be widespread and consistent with other global and regional scale studies, and is based on the fractal properties of landscapes[5,37]. We found that indeed the larger water bodies followed closely the power law ($P < 0.01$) (Supplementary Fig. 1), with each of the ten first-order river basins having unique relationships (Supplementary Fig. 1).

### Estimation of size-dependent trends in land use transitions in the water bodies

We further used the CNLUCC dataset to analyze land use transitions for water bodies smaller than $10^{4.5}$ m$^2$, also referred to as small water bodies (SWB), to other land cover types. To do this each SWB was tracked across the two decades, and their loss pathways to other land use types were quantified. The land use classifications in CNLUCC have been redefined into seven categories: cropland, forest, grassland, construct land, water bodies, saline-alkali land, and other land types. Taking the changes in land use types due to water body loss from 1995 to 2005 as an example, we first overlaid the distributions of small water bodies from both years to identify areas of loss. We then combined this loss distribution with the 1995 land use map to analyze the land use conversion patterns associated with the lost small water bodies.

### N surplus calculations and data sources

The landscape N surplus was calculated for the years 1995, 2005, and 2015 using a soil surface N budget approach[18,39,45]. The basic balance formula for agricultural land is as follows:

$$NS = FERT + MAN + DEP + BNF + STR - CROP \qquad (1)$$

where NS is the N surplus (kg ha$^{-1}$), FERT is fertilizer N application (kg ha$^{-1}$), MAN is the livestock manure returned to the field (kg ha$^{-1}$), DEP is the atmospheric N deposition (kg ha$^{-1}$), BNF is the biological N fixation (kg ha$^{-1}$), STR is the straw return to the field (kg ha$^{-1}$), and is the crop N uptake (kg ha$^{-1}$).

The surplus was estimated at the prefecture-scale, an administrative unit that ranks below provinces and above counties, with 300 such units across China[57]. The methodological details used for our estimation are published in ref. 45, where N surplus estimates were made at the province level, and are briefly described below. The fertilizer data were collected at the prefecture-scale and sourced from statistical yearbooks across different regions, including national statistical yearbooks and provincial-level statistical yearbooks[58]. The national-scale manure use data (MAN) was downscaled to the prefecture-scale based on the percentage of each livestock and poultry category in the prefecture-level city[59]. Atmospheric N deposition data was obtained from ref. 60, which compiled data from the Chinese regional atmospheric deposition network and publicly available N deposition monitoring data in China. Yield-based biological nitrogen fixation was calculated for leguminous crops, while other crops were calculated on the basis of area-based N fixation rates[45]. Straw return to the field was calculated based on the available straw resources and the rate of straw return. The rate at which straw is returned varies by regions and over time[44]. For crop N uptake, we considered 11 crop types (rice, wheat, corn, other cereals, legumes, tuberous crops, oil crops, sugar crops, vegetables, fruits, and other crops), and estimated N uptake as a function of crop yields and crop N content[18,45].

N surplus data at the prefecture scale were downscaled to the 300-m grid scale using the CNLUCC dataset (Supplementary Fig. 5). The CNLUCC dataset was aggregated from 30-m to 300-m resolution based on the dominant land cover designation for each 300-m grid cell. The land within the prefecture-level city is classified into cropland

and non-cropland. The prefecture-scale agricultural N surplus was allocated equally to the cropland grid cells within the prefecture boundaries. Atmospheric N deposition was considered to be evenly distributed across non-cropland grid cells within the prefecture-scale city[36]. The grid-scale surplus data was then further integrated into the third-order river basin (Supplementary Fig. 6). We used three-year averages to mitigate the effects of regional climate variability. Specifically, the nitrogen surplus data for the year 1995 represents the average result for the period 1994 to 1996, and a similar methodology was applied to the results for the years 2005 and 2015.

### Estimating N removal by water bodies

Annual N removal in each of the 258,237 water bodies across China ($R_{wet}$, in kg yr$^{-1}$) was estimated as a function of the corresponding N removal efficiency ($\rho_{wet}$, %) and N inputs ($N_{in}$, in kg yr$^{-1}$), following the method outlined in ref. 36. The N removal efficiencies were estimated through the application of the Hydrobio-k model which was developed based on empirical data from 417 water bodies across China[46]. The HydroBio-k model accounts for both hydrological and biogeochemical processes that influence N removal. It is calibrated using 493 observation data points from China[46]. Given the strong autocorrelation among various hydrological parameters (water depth, area, and residence time), the model with the best calibration results was selected to estimate N removal in water bodies across China. The model assumes first-order reaction kinetics (Eq. 2), with the first-order N removal rate constant ($k$, in d$^{-1}$; Eq. 3) estimated as a function of the residence time ($\tau$, in days), and a temperature correction factor $T_c$ ($= \frac{temperature - 20}{10}$)[45]. The residence time $\tau$ can be estimated as a function of the surface area of the water body (SA, in m$^2$) using an empirical relationship developed (Eq. 4) by ref. 46:

$$\rho_{wet} = (1 - e^{-k\tau}) \times 100 \qquad (2)$$

$$\ln(k) = -1.07 \times \ln(\tau) + 0.29 \times T_c - 0.5 \qquad (3)$$

$$\tau = 1.66 \times SA^{0.24} \qquad (4)$$

Equations (2–4) were used to estimate the N removal efficiencies of each of the 258,237 water bodies across China as a function of their surface area and annual air temperature. The surface area data was obtained from the CNLUCC dataset, and the annual air temperature was obtained from Annual Spatial Interpolation Dataset of Meteorological Elements in China (1-km grid cell)[61]. The actual N removal by a water body ($R_{wet}$, in kg yr$^{-1}$) was then estimated as a function of the N inputs to the water body ($N_{in}$, in kg yr$^{-1}$) and the percent N removal ($\rho_{wet}$).

$$R_{wet} = N_{in} \frac{\rho_{wet}}{100} \qquad (5)$$

Spatially varying N inputs to the water bodies were estimated based on the N surplus of the corresponding third-order river basin ($N_{sur}$, in kg N ha$^{-1}$ yr$^{-1}$), the contributing area of each water body (CA, in ha), and a reduction factor ($\gamma$) that quantifies the fraction of a watershed's N surplus that can enter the water body[36].

$$N_{in} = \gamma N_{sur} CA \qquad (6)$$

The reduction factor $\gamma$ is used to represent the proportion of the N surplus within the catchment area of the water body that has the potential to flow into the wetland, with the rest being retained within the soil or groundwater, or denitrified in upland soils. We assumed $\gamma$ to vary between 0.3 and 0.5 based on the monitoring and modeling-based estimates of soil and water denitrification rates[36]. To capture the regional variations in nitrogen loss across China, $\gamma$ for different

watersheds were estimated based on the proportions of land use types (paddy fields, dryland, forests/grasslands, and constructed areas) and the N loss ratios associated with these land use types. These parameters were then normalized to a range of 0.3–0.5:

$$\gamma = \left( \sum_{i=1}^{4} f_i \times LR_i \right) \times 0.2 + 0.3 \qquad (7)$$

where $i$ represents different land use types, including paddy fields, dryland, forests/grassland, and constructed land; $f$ represents the proportions of land use types at different third-order basins; $LR$ represents the N loss ratios associated with these land use types. For paddy fields and drylands, it includes distinct nitrogen loss rates from different nitrogen sources, such as synthetic fertilizers and livestock manure. These data are derived from monitoring and simulation reported in the previous study[18], with parameter details listed in Supplementary Table 3. The $\gamma$ for each watershed are shown in Supplementary Fig. 7. Based on existing research on nitrogen loss, we assumed that the $\gamma$ follows a normal distribution with a 25% coefficient of variation to account for uncertainty[62]. The contributing area of each water body (CA) was estimated by assuming a watershed-area-to-water body area ratio of $\alpha$. This parameter is an empirical value derived from the water body catchment area and water body surface area, with the range of values based on empirical data[63]. We used the geometric mean and standard deviation of the data to bound this parameter in our Monte Carlo simulations[37]. Future refinements to this methodology will focus on delineating the watersheds of all waterbodies across China. This will become achievable as high-resolution data becomes increasingly available for additional regions. In regions with a large density of water bodies, the sum of the contributing areas can sometimes be greater than the basin area. In these cases, the $\alpha$ was rescaled to make sure the sum of water body contributing area (CA) is equal to the basin area, following ref. 36. This recalibration was performed to ensure that the model calculates that the mass of N entering the water body is not greater than the mass of N available within the basins. Finally, the mass removal at the watershed-scale ($M_{wshd}$, in kg ha$^{-1}$ yr$^{-1}$; Fig. 4a) was estimated by calculating the N removal of all water bodies within each third-order river basins, and then normalizing by the basin area:

$$M_{wshd} = \frac{\sum_{i=1}^{n} R_{wet,i}}{A_{shd}} \qquad (8)$$

where $R_{wet,i}$ is N removal by an individual water body (kg), $n$ is the number of water bodies within the basin, $A_{shd}$ is the basin area (ha). Monte Carlo simulations were used to consider the uncertainty of the parameters $\alpha$ and $\gamma$, which were found to be the most sensitive parameters based on a previous analysis[46]. The range of parameters chosen for $\gamma$ and $\alpha$ was 0.3–0.5 and 3–20, respectively[36]. We conducted 1,000 Monte Carlo simulations by randomly drawing from the distribution of these parameters. The estimation of median and interquartile ranges (IQR) of N removal are provided in the main text.

### Wetland restoration scenarios

We simulated three different wetland restoration scenarios at the first-order river basin scale to understand the role of water body size in its N removal potential. Total area restored was maintained as the same across all three scenarios, but different in the different basins, and estimated as a function of spatially varying wetland loss patterns (Supplementary Table 4). Specifically, we assumed that wetland restoration was used to recover the theoretical fractal distribution that was disrupted by the loss of the smaller water bodies (shown as gray bars in Supplementary Fig. 1). The power-law distribution of water-bodies has been applied in existing studies to estimate their theoretical distribution[64] and potential loss[26]. The percent water body area that

had to be restored ranged from 1 to 36% of the current area of the water bodies in the corresponding basin (Fig. 4b, Supplementary Table 4).

In Scenario 1 (S1), we assumed that we restored smaller water bodies of varied sizes but with an area less than $10^{4.5}$ m$^2$, while in the second (S2) and the third scenarios (S3) we assumed the area restored to be the same as S1, but the size of the restored water bodies to be larger. Specifically, in S2 we assumed the overall area restoration to be achieved by restoring ten water bodies of equal area, and in S3 we assumed that only one very large water body was restored. The wetland areas corresponding to the three scenarios are provided in Supplementary Table 4. Our choice of a constant threshold of $10^{4.5}$ m$^2$ across the country for estimating the restored area is an assumption, which is justified given our objective was to test whether restoring water bodies of different sizes have an impact on nutrient retention. A different threshold value would lead to different magnitudes of area restored, but the comparison between small water bodies and large water bodies would still be valid.

In S1, the restored wetlands are assumed to be distributed across the first-order basins, following their original distribution along the fractal line; The location question is a bit more tricky for scenarios 2 and 3, given larger wetlands cannot be placed everywhere at the same time. To address the location uncertainty, we ran multiple scenarios assuming that the ten wetlands in S2 and the one wetland in S1 can be placed in any of the third-order basins nested within the corresponding first-order basins.

The N removed by the restored wetlands was then estimated as a function of their size that drives their residence times and their locations that determine the N surplus and temperature factors (Eq. 3). In S1, we assume that the restored wetlands have the same temperature and N surplus distribution as the existing wetlands in the corresponding first-order watershed (i.e. the orange bar in Supplementary Fig. 1).

Finally, parameter uncertainty was considered for all three scenarios by doing 1000 Monte Carlo simulations using the range of model parameters as described above[36]. The median and interquartile range values were reported in the main text, illustrating the uncertainty associated with water body restoration at various locations and parameter uncertainty (Fig. 4b).

### Estimation of economic value of restoration

Small water bodies provide multiple ecosystem services such as protection from flooding, water purification, habitat and carbon sequestration. Here, we provide an estimate for the economic value of water purification ($E_w$) using data from wastewater treatment plants (WWTPs),

$$E_w = \frac{R_{awet}}{C_{in} - C_{out}} \times T_{cost} \times 1000 \tag{9}$$

where $R_{awet}$ (kg yr$^{-1}$) is N removal by all water bodies across China, $C_{in}$ (in mg L$^{-1}$) is the mean influent TN concentration of WWTPs in China during 2015, $C_{out}$ (in mg L$^{-1}$) is the mean effluent TN concentration of WWTPs in China during 2015 (ref. 50), $T_{cost}$ (CNY m$^{-3}$) is the water treatment cost by WWTPs[51]. The exchange rate between RMB and USD is calculated based on 2015. Inflation is ignored in exchange rate conversion calculations. Our calculation method uses WWTPs as a reference. While effective for comparison, it overlooks the broader value water bodies offer beyond N reduction, such as their landscape and ecological contributions. Therefore, this approach underestimates the full value of water bodies.

### Data availability

All data used in this study are available from the sources listed here. Land use data (CNLUCC) were obtained from the Data Center for Resources and Environmental Sciences, Chinese Academy of Sciences (https://www.resdc.cn/). The watershed boundary was obtained from National earth system science data center (http://lake.geodata.cn). The data used for N surplus calculation was retrieved from regional and national statistical yearbooks (https://data.stats.gov.cn/easyquery.htm?cn=C01), and the Food and Agriculture Organization database (https://www.fao.org/faostat/en/#data). The data generated in this study are provided in the Supplementary Information and Source Data file. Source data are provided with this paper.

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

## Acknowledgements

This work was supported by the National Natural Science Foundation of China (U21A2025, L.Z. and 42207059, S.L.), the National Key Research and Development Program of China (2022YFD1700700, L.Z.), Hubei

Provincial Natural Science Foundation of China (2024AFA020, L.Z.), and the China Scholarship Council (202204910377, W.S.).

## Author contributions

W.S., L.Z., and N.B.B. designed the research. W.S. and B.X. collected the data. W.S. analyzed the data and created the figures with advice from L.Z., E.A.U., S.L., and N.B.B. W.S. and N.B.B. led the writing of the paper with substantial input from E.A.U. and L.Z.

## Competing interests

The authors declare no competing interests.
