## [Peer Review file · Nature Communications]

Restoring small water bodies to improve lake and river water quality in China

Corresponding Author: Professor Liang Zhang

Version 0:

Reviewer comments:

Reviewer #1

(Remarks to the Author)

Lentic systems are critical resources that play major roles in global hydrologic and biogeochemical cycles, while they are also suffering from increasing pressures from climate change and intensified anthropogenic activities. The manuscript raises an interesting topic on improving lake and river water quality in China by restoring small water bodies. The authors utilize land use from 1995, 2005, and 2015 to identify spatial patterns of small water body loss, calculate nitrogen removal, identify high-risk regions for water quality based on the reduction of small water bodies and cropland nitrogen surplus, and assess the value of nitrogen removal. However, the current manuscript does not provide enough reliable and robust analysis to support the conclusions.

1. The area of water bodies is critical for this study; however, the authors did not scientifically and reasonably define the classification of water bodies. Specifically:

(1) How are small water bodies defined? Figures 1b and 1c do not provide a scientific basis for this classification.

(2) In the restoration scenarios, S1 involves restoring small water bodies, S2 involves restoring 10 larger water bodies, and S3 involves restoring one very large water body. How are the sizes of water bodies in these three scenarios defined? Where are they located? How are related parameters such as elevation, depth, volume, and hydraulic residence time determined? All these factors can affect nitrogen removal efficiency and need to be clarified first.

2. The nitrogen flux into water bodies is the most critical variable in this study, and there are significant scientific flaws in its calculation. Specifically:

(1) The entire model system does not use water quality monitoring data/observations to validate/justify the accuracy of the model, and it also lacks explanations for the rationality of model parameters.

(2) In lines 136-137, "To identify high-risk areas for water quality degradation, we used a recently developed map of grid-scale, landscape nitrogen surplus across China (Fig. 3b)," while the manuscript does not provide the grid layer for nitrogen surplus.

(3) The nitrogen entering water bodies in a watershed is related to factors such as rainfall, slope, and soil properties. The use of function ($N_{in} = \gamma N_{surCA}$) to calculate nitrogen input is fundamentally flawed. China's vast territory spans five climatic zones with significant variations in topography, making it impossible for the constant γ to represent the nitrogen input ratio for all lakes in 1995, 2005, and 2015.

(4) Defining high-risk areas using the 25th percentile of nitrogen surplus is an overestimate, as China's nitrogen use efficiency is lower than the global average. It is recommended to use the global average as the basis for definition and to refine the values for China.

(5) The study does not define or explain "future high-risk" regions. Throughout the manuscript, the authors do not mention future climate or economic development scenarios, nor do they calculate future nitrogen surplus and changes in water bodies.

(6) In lines 439 to 441, the authors downscale national data to the prefecture level, and in line 451, they downscale prefecture-level data to a 300-meter grid, without explaining the specific downscaling methods, reliability and uncertainties.

3. In Figure 1d and Extended Data Figure 1, why not using a one-to-one correspondence method to calculate the loss of small water bodies, but instead defining the loss of small water bodies as the number not reaching the power function regression? Extended Data Figure 1, especially for SRB and LRB, why are the blank areas not defined as potential losses with the definition provided in Figure 1d?

Reviewer #2

(Remarks to the Author)

Reviewer #3

(Remarks to the Author)

This manuscript focuses on the loss of small water bodies in China and their potential impact on water quality. This topic is certainly of interest to a broader audience, as many regions worldwide are grappling with similar challenges. Although the importance of small water bodies has already been widely recognized, this manuscript provides a compelling demonstration that restoring small water bodies can bring greater environmental benefits than restoring large water bodies. The findings of this paper not only highlight the challenge China is facing with the loss of small water bodies but also strongly support strategies for water body restoration. In my view, this paper is well-written and answers important research questions. I encourage authors to consider the following points to improve their article further.

(1) The meanings of “current high-risk” and “future high-risk” need to be further clarified.

(2) For the convenience of potential readers to conduct comparative studies with other regions, the annual reduction rates of small water bodies in national scale and key regions should be quantified respectively.

(3) As not all readers are familiar with the characteristics of different basins, I recommend authors provide more information about 10 first-class river basins in the extended data to help explain the spatial results.

Version 1:

Reviewer comments:

Reviewer #1

(Remarks to the Author)

Lentic water bodies are a crucial component of terrestrial ecosystems, playing a vital role in nitrogen removal. The manuscript provides a systematic evaluation of the nitrogen removal potential of water bodies across China, emphasizing the importance of restoring small water bodies. While previous research has underscored the importance of these smaller water systems, this study goes further by offering a compelling visual demonstration of their restoration benefits, illustrating their capacity to improve water quality. In wetland restoration practices, the debate continues on whether to prioritize restoring large wetlands or multiple smaller ones. This study offers strong evidence from a water quality protection perspective, providing a solid foundation for policymaking. I believe this study can draw widespread attention.

I read through the comments and the responses that the authors provide. Generally, the authors have enhanced the methodology by adding more details and accounting for regional differences across China, which strengthens the study's reliability. Although there is still room to improve the methods for calculating nitrogen inputs in wetlands, the current approach aligns with the study's objectives and provides a basis for evaluating nitrogen dynamics in wetlands. I think the current method is acceptable.

Some more suggestions to improve:

(1) The authors employed a power-law distribution to estimate water bodies loss before 2015. This method is new to me; has it been used in previous research? I suggest that the authors offer a more detailed explanation of this approach.

(2) The authors used wastewater treatment plants to estimate the economic benefits of wetland restoration, which is fine. However, the limitations of this method should be further clarified.

(3) In the revisions, the authors have already addressed the national standards for small wetlands. I suggest that the authors involved more content related to wetland policies, which would further enhance the study's impact on policy development.

Reviewer #3

(Remarks to the Author)

This manuscript focuses on the interplay of nitrogen inputs and inland water body dynamics, and the implications for water quality at a national scale. The perspective and new findings of this manuscript are impressive. This manuscript emphasizes the size matters and location matters of water bodies, and underscores the importance of protecting and restoring small water bodies as vital components of the landscape, not only for their role in global hydrologic and biogeochemical cycles but also for their significant contribution to water quality improvement. In the past 20 years, China has carried out extensive government guided and multi-party participation in the construction of constructed wetlands and wastewater treatment infrastructure, as well as the restoration of rivers, lakes, and reservoirs, to alleviate the pressure of climate change and water quality deterioration. The investment in pollution control and water body restoration in China was enormous, and many developing countries around the world are also facing the similar challenges. So, in this context, this study is particularly important. Several issues regarding description and method have been addressed. Overall, I think that the current version can be accepted for publication.

Reviewer #1

Lentic systems are critical resources that play major roles in global hydrologic and biogeochemical cycles, while they are also suffering from increasing pressures from climate change and intensified anthropogenic activities. The manuscript raises an interesting topic on improving lake and river water quality in China by restoring small water bodies. The authors utilize land use from 1995, 2005, and 2015 to identify spatial patterns of small water body loss, calculate nitrogen removal, identify high-risk regions for water quality based on the reduction of small water bodies and cropland nitrogen surplus, and assess the value of nitrogen removal. However, the current manuscript does not provide enough reliable and robust analysis to support the conclusions.

Response: Thanks for your comments. Your insightful suggestions have been very helpful in improving the paper. We responded to your suggestions accordingly.

1. The area of water bodies is critical for this study; however, the authors did not scientifically and reasonably define the classification of water bodies. Specifically:

(1) How are small water bodies defined? Figures 1b and 1c do not provide a scientific basis for this classification.

Response: In this paper, small water bodies are defined as those with an area less than $10^{4.5}$ m² (Lines 82-84). This definition conforms with the Chinese national standard “*Specification for conservation and management of small wetlands*” (GB/T 42481-2023), though we opted to use the broader term “water bodies”. We revised the manuscript to clarify and cite this standard (Lines 82-84, 458-460). Also note that in Figures 1b and 1c, the % loss for the small and large water bodies turn out as very different; and we explore loss across different size classes and not only at the threshold.

(2) In the restoration scenarios, S1 involves restoring small water bodies, S2 involves restoring 10 larger water bodies, and S3 involves restoring one very large water body. How are the sizes of water bodies in these three scenarios defined? Where are they located? How are related parameters such as elevation, depth, volume, and hydraulic residence time determined? All these factors can affect nitrogen removal efficiency and need to be clarified first.

Response: Thanks for your comments. We describe below how the sizes were defined, and then the sizes were used to estimate the residence times following the empirical equations we developed in our earlier work. We added more details in the Methods and Supplementary materials to clarify the method we used:

“We simulated three different wetland restoration scenarios at the first-order river basin scale to understand the role of water body size in its N removal potential. Total area restored was maintained as the same across all three scenarios, but different in the different basins, and estimated as a function of spatially varying wetland loss patterns (Supplementary Table 4). Specifically, we assumed that wetland restoration was used to recover the theoretical fractal distribution that was

disrupted by the loss of the smaller water bodies (shown as gray bars in Supplementary Fig. 1). The percent water body area that had to be restored ranged from 1 to 36% of the current area of the water bodies in the corresponding basin (Fig. 4b, Supplementary Table 4).

In Scenario 1 (S1), we assumed that we restored smaller water bodies of varied sizes but with an area less than $10^{4.5}$ m², while in the second (S2) and the third scenarios (S3) we assumed the area restored to be the same as S1, but the size of the restored water bodies to be larger. Specifically, in S2 we assumed the overall area restoration to be achieved by restoring ten water bodies of equal area, and in S3 we assumed that only one very large water body was restored. The wetland areas corresponding to the three scenarios are provided in Supplementary Table 4. Our choice of a constant threshold of $10^{4.5}$ m² across the country for estimating the restored area is an assumption, which is justified given our objective was to test whether restoring water bodies of different sizes have an impact on nutrient retention. A different threshold value would lead to different magnitudes of area restored, but the comparison between small water bodies and large water bodies would still be valid.

In S1, the restored wetlands are assumed to be distributed across the first-order basins, following their original distribution along the fractal line; The location question is a bit more tricky for scenarios 2 and 3, given larger wetlands cannot be placed everywhere at the same time. To address the location uncertainty, we ran multiple scenarios assuming that the ten wetlands in S2 and the one wetland in S1 can be placed in any of the third order basins nested within the corresponding first order basins.

The nitrogen removed by the restored wetlands was then estimated as a function of their size that drives their residence times and their locations that determine the nitrogen surplus and temperature factors (Eq 3). We assume that the restored wetlands have the same temperature and N surplus distribution as the existing wetlands in the corresponding first-order watershed (i.e. the orange bar in Supplementary Fig. 1).” (Lines 589-629)

2. The nitrogen flux into water bodies is the most critical variable in this study, and there are significant scientific flaws in its calculation. Specifically:

(1) The entire model system does not use water quality monitoring data/observations to validate/justify the accuracy of the model, and it also lacks explanations for the rationality of model parameters.

Response: Thank you for your comments. We agree that the nitrogen flux into the water bodies is a critical variable. However, the challenge is that water quality monitoring data that is available in China and globally, are at the watershed outlet. Such water quality monitoring data in streams cannot be used for models such as this, given those concentrations are available at larger scales, where high nitrogen

concentrations that enter small wetlands have already been diluted. One can potentially develop a watershed model with hundreds of wetlands and calibrate the model to measured concentrations at the outlet, but such models have hundreds of calibration parameters, and thus uncovering the role of the wetlands is impossible. Also, the method is not transferable to a larger scale analysis.

Our method to estimate nitrogen flux into wetlands can definitely be improved, but currently data doesn't exist at such large scales for these estimates. Most previous estimates of wetland nutrient retention at large scales have not taken into spatially varying nitrogen inputs as a function of spatially varying N surplus. The method was developed in a recent paper by Cheng et al., (2020, *Nature*) who showed the value of the methodology over existing approaches.

To further clarify the model parameters used in our text, we added more details of the model in our Method. (Lines 522-531)

Reference: Cheng, F. Y., et al. Maximizing US nitrate removal through wetland protection and restoration. *Nature*, 2020, 588, 625-630.

(2) In lines 136-137, "To identify high-risk areas for water quality degradation, we used a recently developed map of grid-scale, landscape nitrogen surplus across China (Fig. 3b)," while the manuscript does not provide the grid layer for nitrogen surplus.

Response: Thanks for your comments. We added the grid-scale nitrogen surplus data in the Supplementary Information.

Supplementary Fig. 6. Grid-scale nitrogen surplus data across China in 2015.

(3) The nitrogen entering water bodies in a watershed is related to factors such as rainfall, slope, and soil properties. The use of function to calculate nitrogen input is fundamentally flawed. China's vast territory spans five climatic zones with significant variations in topography, making it impossible for the constant to represent the nitrogen input ratio for all lakes in 1995, 2005, and 2015.

Response: In this article, we used nitrogen surplus and reduction factor (γ) to estimate nitrogen inputs into water bodies. We also used Monte Carlo to assess the uncertainty in our estimates based on uncertainty in the reduction factor. As you correctly note, there is significant uncertainty in this factor as a function of climate and landscapes attributes. Developing a model to accurately assess spatially varying gamma would require plot-scale nitrogen leaching data across the country that is currently unavailable to the best of our knowledge.

As you mentioned, China's regions display significant spatial heterogeneity. We improved our method to better account for the differences in nitrogen loss across various regions in China. The details are as follows:

“We assumed γ to vary between 0.3 and 0.5 based on the monitoring and modeling based estimates of soil and water denitrification rates (Cheng et al., 2020, Nature). To capture the regional variations in nitrogen loss across China, γ for different watersheds were estimated based on the proportions of land use types (paddy fields, dryland, forests/grasslands, and constructed areas) and the N loss ratios associated with these land use types. These parameters were then normalized to a range of 0.3-0.5:

$$\gamma = \left(\sum_{i=1}^4 f_i \times LR_i \right) \times 0.2 + 0.3$$

where i represents different land use types, including paddy fields, dryland, forests/grassland, and constructed land; f represents the proportions of land use types at different third-order basins; LR represents the N loss ratios associated with these land use types. For paddy fields and drylands, it includes distinct nitrogen loss rates from different nitrogen sources, such as synthetic fertilizers and livestock manure. These data are derived from monitoring and simulation reported in the previous study (Gu et al., 2015, PNAS), with parameter details listed in Supplementary Table 3. The γ for each watershed are shown in Supplementary Figure 7. Based on existing research on nitrogen loss, we assumed that the γ follows a normal distribution with a 25% coefficient of variation to account for uncertainty (Zhang et al., 2021, EP).” (Lines 551-556)

Supplementary Fig. 7. Reduction factor at different watersheds.

Reference: Cheng, F. Y., et al. Maximizing US nitrate removal through wetland protection and restoration. *Nature*, 2020, 588, 625-630.

Gu, B. J., et al. Integrated reactive nitrogen budgets and future trends in China. *Proc. Natl. Acad. Sci. U.S.A.*, 2015, 112, 8792-8797.

Zhang, X. M., Ren, C. C., Gu, B. J. & Chen, D. L. Uncertainty of nitrogen budget in China. *Environ. Pollut.* 286, 117216.

(4) Defining high-risk areas using the 25th percentile of nitrogen surplus is an overestimate, as China's nitrogen use efficiency is lower than the global average. It is recommended to use the global average as the basis for definition and to refine the values for China.

Response: Thanks for your suggestion. We revised the manuscript to use the global average nitrogen surplus (Zhang et al., 2021, *Nature Food*) as the basis for defining high-risk areas (Lines 159-161, Figure 3).

Reference: Zhang, X., et al. Quantification of global and national nitrogen budgets for crop production. *Nature Food*, 2021, 2(7), 529-540.

(5) The study does not define or explain "future high-risk" regions. Throughout the manuscript, the authors do not mention future climate or economic development scenarios, nor do they calculate future nitrogen surplus and changes in water bodies.

Response: Thanks for your comments. We did define "future high-risk areas" as below:

"We also identified "future high-risk" regions (red region in Fig. 3c) as areas where N surplus is increasing accompanied by a loss of small water bodies"

But, we realize that the choice of the terms was not clarified adequately. We have now added the following text to address this. Note, future projections for N surplus changes or land use transitions that would contribute to loss of small water bodies require land use modelling that is beyond the scope of our study.

“We use the temporal trends in our N surplus and small water body datasets to identify areas that might be at greater risk to nitrogen pollution. We defined “current high-risk” regions as those with a high N surplus and where small bodies are being drained in the last two decades – these are intensive agricultural regions where wetlands are needed to tackle the high N surplus. We further define “future high-risk” areas as those where the N surplus is increasing while SWB are decreasing from 1995 to 2015. These are potentially less agriculturally intensive regions, where agricultural activities and thus N surplus is increasing, while small water bodies are being lost. Their N surplus magnitudes might still be low, but the increasing trend of N sources, with concomitant decrease in small water bodies create a high risk of future water quality deterioration.” (Lines 149-158)

(6) In lines 439 to 441, the authors downscale national data to the prefecture level, and in line 451, they downscale prefecture-level data to a 300-meter grid, without explaining the specific downscaling methods, reliability and uncertainties.

Response: Thanks for your comments. We revised the manuscript and added extended figures to explain the scale transformation process (Lines 507-514, Supplementary Fig. 5).

In this paper, the scale transformation of nitrogen surplus from the prefecture-level city scale to the watershed scale is divided into two steps. The first step involves transforming the prefecture-level city into a grid scale. Based on land use type data, the land within the prefecture-level city is classified into cropland and non-cropland. The nitrogen surplus for cropland is calculated based on the nitrogen budget (Equation 1), while the nitrogen surplus for non-cropland is assigned using atmospheric deposition values. The second step involves integrating the grid-scale nitrogen surplus into the watershed scale, where the watershed-scale nitrogen surplus is the average of the grid values within the watershed.

Supplementary Fig. 5. The scale transformation process of nitrogen surplus.

3. In Figure 1d and Extended Data Figure 1, why not using a one-to-one correspondence

method to calculate the loss of small water bodies, but instead defining the loss of small water bodies as the number not reaching the power function regression? Supplementary Figure 1, especially for SRB and LRB, why are the blank areas not defined as potential losses with the definition provided in Figure 1d?

Response: Thanks for your comments. In Figure 1a, b and c we do use a point to point comparison to estimate the loss of water bodies as a function of their size classes. We realize, however that the writing was confusing and have thus edited to say this more clearly.

The goal of Supplementary Figure 1 is to show how we created the restoration scenarios. We wanted to create restoration scenarios that considered loss that occurred even before 1995. To do this, we used the power law breakpoint to recreate the original theoretical power law function. You are correct in noting that because this breakpoint occurred at different sizes in different basins – for example, it occurred at larger sizes in the SRB and LRB – resulting in some white areas not included in our scenario analysis. However, setting different thresholds for each first-order basin would make the comparison across them to be challenging. The goal of our scenario analysis was to demonstrate that restoring small versus large wetlands contributes to different outcomes even when the same area is restored, and we believe this approach meets that goal. The 7% increase in wetland area for our three scenarios is of course a function of the threshold of $10^{4.5}$ m² selected for the study. While a different threshold value would lead to a different value for the area restored, our main finding that restoring smaller water bodies would have much greater water quality benefits would still be valid. (Lines 236-239)

Reviewer #2 (Remarks to the Author):

Response: Thanks.

Reviewer #3 (Remarks to the Author):

This manuscript focuses on the loss of small water bodies in China and their potential impact on water quality. This topic is certainly of interest to a broader audience, as many regions worldwide are grappling with similar challenges. Although the importance of small water bodies has already been widely recognized, this manuscript provides a compelling demonstration that restoring small water bodies can bring greater environmental benefits than restoring large water bodies. The findings of this paper not

only highlight the challenge China is facing with the loss of small water bodies but also strongly support strategies for water body restoration. In my view, this paper is well-written and answers important research questions. I encourage authors to consider the following points to improve their article further.

Response: Thanks for your comments and kind words. Your suggestions have been very helpful in improving the paper. We responded to your suggestions accordingly.

(1) The meanings of “current high-risk” and “future high-risk” need to be further clarified.

Response: Thanks for your comments. We revised the manuscript to clarify the description of current and future high-risk regions. (Lines 149-158).

“We use the temporal trends in our N surplus and small water body datasets to identify areas that might be at greater risk to nitrogen pollution. We defined “current high-risk” regions as those with a high N surplus and where small bodies are being drained in the last two decades – these are intensive agricultural regions where wetlands are needed to tackle the high N surplus. We further define “future high-risk” areas as those where the N surplus is increasing while SWB are decreasing from 1995 to 2015. These are potentially less agriculturally intensive regions, where agricultural activities and thus N surplus is increasing, while small water bodies are being lost. Their N surplus magnitudes might still be low, but the increasing trend of N sources, with concomitant decrease in small water bodies create a high risk of future water quality deterioration.” (Lines 149-158)

(2) For the convenience of potential readers to conduct comparative studies with other regions, the annual reduction rates of small water bodies in national scale and key regions should be quantified respectively.

Response: Thanks for your comments. We revised the manuscript to show the annual reduction rates of small water bodies (Supplementary Table 1).

Supplementary Table 1 The annual change in the number and area of small water bodies

Basin	The average change in the number of small water bodies (yr ⁻¹)	The average change in the area of small water bodies (km ² yr ⁻¹)
Songhua River Basin	-1041	-2.3
Liaohe River Basin	-159	-0.2
Haihe River Basin	-59	0.8
Yellow River Basin	-1783	-3.9
Huaihe River Basin	-69	1.3
Yangtze River Basin	-3680	-12.6
Southeast Basin	-4	-0.3
Pearl River Basin	-218	-0.2
Southwest Basin	-371	-1.4
Continental Basin	-2612	-5.9

(3) As not all readers are familiar with the characteristics of different basins, I recommend authors provide more information about 10 first-class river basins in the Supplementary to help explain the spatial results.

Response: Thanks for your suggestions. We provided more information about first-class river basins in the Supplementary (Supplementary Table 2).

Reviewer #1 (Remarks to the Author):

Lentic water bodies are a crucial component of terrestrial ecosystems, playing a vital role in nitrogen removal. The manuscript provides a systematic evaluation of the nitrogen removal potential of water bodies across China, emphasizing the importance of restoring small water bodies. While previous research has underscored the importance of these smaller water systems, this study goes further by offering a compelling visual demonstration of their restoration benefits, illustrating their capacity to improve water quality. In wetland restoration practices, the debate continues on whether to prioritize restoring large wetlands or multiple smaller ones. This study offers strong evidence from a water quality protection perspective, providing a solid foundation for policymaking. I believe this study can draw widespread attention.

I read through the comments and the responses that the authors provide. Generally, the authors have enhanced the methodology by adding more details and accounting for regional differences across China, which strengthens the study's reliability. Although there is still room to improve the methods for calculating nitrogen inputs in wetlands, the current approach aligns with the study's objectives and provides a basis for evaluating nitrogen dynamics in wetlands. I think the current method is acceptable.

Response: Thanks for your comments and kind words. Your suggestions have been very helpful in improving the paper. We responded to your suggestions accordingly.

Some more suggestions to improve:

(1) The authors employed a power-law distribution to estimate water bodies loss before 2015. This method is new to me; has it been used in previous research? I suggest that the authors offer a more detailed explanation of this approach.

Response: The power-law distribution of water body areas and sizes has been extensively validated across various scales in existing studies, such as Van Meter & Basu (2015), Downing (2010), and Sagar (2007). Furthermore, this distribution has also been applied to estimate the number of waterbodies and potential waterbody loss (e.g., Downing et al., 2006; Serran & Creed, 2016).

We revised the manuscript and incorporated additional references to clarify the approach based on the power-law distribution (Lines 461, 592-593).

“We then fitted a power function relationship to water bodies larger than this threshold, following the assumption that preferential loss of small water bodies creates deviations from the theoretical power function relationships between the size of the water bodies and their abundance across the landscape ($N_w = cA^d$; A in m^2 is the surface area of the water body, N_w is the number of water bodies corresponding to the size, and c and d are empirical constants). The power-law relationship is considered to be widespread and consistent with other global and regional scale studies, and is based on the fractal properties of landscapes.” (Lines 456-465)

“Specifically, we assumed that wetland restoration was used to recover the theoretical fractal distribution that was disrupted by the loss of the smaller water bodies (shown as gray bars in Supplementary Fig. 1). The power-law distribution of waterbodies has been applied in existing studies to estimate their theoretical distribution and potential loss.” (Lines 590-593)

References:

Van Meter, K. J. & Basu, N. B. (2015) Signatures of human impact: size distributions and spatial organization of wetlands in the Prairie Pothole landscape. *Ecological Applications*, 25, 451-465.

Downing, J. A. (2010) Emerging global role of small lakes and ponds: little things mean a lot. *Limnetica* 29, 9-23.

Sagar, B. D. (2007). Universal scaling laws in surface water bodies and their zones of influence. *Water Resources Research*, 43(2).

Downing, J. A., et al. (2006). The global abundance and size distribution of lakes, ponds, and impoundments. *Limnology and Oceanography*, 51(5), 2388-2397.

Serran, J. N., & Creed, I. F. (2016). New mapping techniques to estimate the preferential loss of small wetlands on prairie landscapes. *Hydrological Processes*, 30(3), 396-409.

(2) The authors used wastewater treatment plants to estimate the economic benefits of wetland restoration, which is fine. However, the limitations of this method should be further clarified.

Response: Thanks for your suggestion. Overall, using wastewater treatment plants as a reference for evaluating the economic value of water bodies may undervalue their true significance. Compared to wastewater treatment plants, water bodies offer additional benefits, such as landscape and ecological value. We have revised the methodology section to more clearly explain the limitations of this approach. (Lines 633-636)

“Our calculation method uses wastewater treatment plants as a reference. While effective for comparison, it overlooks the broader value water bodies offer beyond N reduction, such as their landscape and ecological contributions. Therefore, this approach underestimates the full value of water bodies.” (Lines 633-636)

(3) In the revisions, the authors have already addressed the national standards for small wetlands. I suggest that the authors involved more content related to wetland policies, which would further enhance the study’s impact on policy development.

Response: Thanks for your suggestion. We revised the both Introduction and Results parts to involve more wetland-related policies to enhance the study’s impact on policy development. (Lines 59-64, 235-241)

“Since joining the Ramsar Convention on Wetlands in 1992, China has advanced

wetland conservation through efforts like the 2003 National Wetland Conservation Program, which aimed to protect 90% of natural wetlands and restore 1.4 million hectares. At the 2022 Ramsar Conference (COP14), China proposed a resolution to protect and restore small wetlands, urging global action. However, the environmental benefits of restoring small wetlands, compared to large ones, remain unclear” (Lines 59-64)

“Traditional wetland conservation often prioritizes large wetlands, with the 82 Ramsar-listed wetlands in China averaging 100,000 hectares. However, our simulation results underscore the vital role of small wetlands in pollution control. This finding strongly supports China’s National Wetland Conservation Program, the fulfillment of responsibilities by the Ramsar Convention on Wetlands’ Contracting Parties worldwide, and the pursuit of Sustainable Development Goals through effective wetland restoration.” (Lines 235-241)

Reviewer #3 (Remarks to the Author):

This manuscript focuses on the interplay of nitrogen inputs and inland water body dynamics, and the implications for water quality at a national scale. The perspective and new findings of this manuscript are impressive. This manuscript emphasizes the size matters and location matters of water bodies, and underscores the importance of protecting and restoring small water bodies as vital components of the landscape, not only for their role in global hydrologic and biogeochemical cycles but also for their significant contribution to water quality improvement. In the past 20 years, China has carried out extensive government guided and multi-party participation in the construction of constructed wetlands and wastewater treatment infrastructure, as well as the restoration of rivers, lakes, and reservoirs, to alleviate the pressure of climate change and water quality deterioration. The investment in pollution control and water body restoration in China was enormous, and many developing countries around the world are also facing the similar challenges. So, in this context, this study is particularly important. Several issues regarding description and method have been addressed. Overall, I think that the current version can be accepted for publication.

Response: Thanks.